# Minimally invasive surgery and neurophysiological monitoring for brainstem hemorrhage: Advancing predictive models with qEEG and TCD

Jin Wang[1], Huayong Wu[2], Bingjie Jiang [2*]

**1** Department of Surgery, The Quzhou Affiliated Hospital of Wenzhou Medical University, Quzhou People's Hospital, Quzhou, People's Republic of China, **2** Department of Neurosurgery, The Quzhou Affiliated Hospital of Wenzhou Medical University, Quzhou People's Hospital, Quzhou, People's Republic of China

* jiangbingjie123@126.com

## Abstract

### Background

Primary brainstem hemorrhage (PBSH) is a life-threatening neurological condition associated with high mortality and disability rates. Stereotactic hematoma aspiration surgery has been explored as a treatment option, and postoperative brainstem function monitoring is considered important for patient management.

### Objective

This study aimed to evaluate the integration of minimally invasive stereotactic aspiration surgery with quantitative electroencephalography (qEEG) and transcranial Doppler (TCD) monitoring to assess brain function and improve predictive models for clinical outcomes in PBSH patients.

### Methods

We conducted a retrospective analysis of 34 PBSH patients admitted between December 2022 and October 2023. After applying exclusion criteria, 25 eligible patients underwent stereotactic aspiration surgery within 24–48 hours of symptom onset. Both qEEG and TCD monitoring were performed preoperatively and within 24 hours postoperatively. Changes in qEEG parameters and TCD-derived hemodynamic indices were analyzed to assess surgical safety and efficacy.

### Results

Stereotactic surgery was associated with higher rates of favorable outcomes at 90 days compared with the non-surgical group (68.75% vs. 11.11%, p = 0.01). Postoperative TCD parameters improved significantly, indicating better hemodynamic stability,

**Data availability statement:** All relevant data are available from Figshare at https://doi.org/10.6084/m9.figshare.30018478.

**Funding:** This work was supported by the Key Scientific and Technological Project of Zhejiang Province (Quzhou City Guided Project; Grant Nos. 2021022 and 2018086) and the Zhejiang Medical Association Clinical Research Fund Project (Grant No. 2022ZTC-A111). There was no additional external funding received for this study. The funder had the following involvement with the study: study design, decision to publish, and preparation of the manuscript.

**Competing interests:** The authors have declared that no competing interests exist.

**Abbreviations:** PBSH, primary brainstem hemorrhage; qEEG, quantitative electroencephalography; TCD, transcranial Doppler; mRS, Modified Rankin Scale; PI, Pulsatility index; VM, mean flow velocity; GCS, Glasgow Coma Scale; RBP, relative band power; VS, systolic velocity; VD, diastolic velocity; VM, mean velocity; SD, standard deviation; IQR, interquartile range; ROC, receiver operating characteristic; AUC, area under the curve; CT, computed tomography; aEEG, amplitude-integrated electroencephalogram.

though no correlation with mRS scores was found. qEEG analysis showed significant correlations between RBP $\delta$% and mRS scores ($\rho = 0.480$, p = 0.015), and RBP $\alpha$% ($\rho = -0.456$, p = 0.022). aEEG also correlated strongly with 90-day mRS scores ($\rho = 0.544$, p = 0.004). The combined model of hematoma volume, RBP $\alpha$%, and aEEG showed the highest predictive accuracy (AUC = 0.865).

## Conclusion

This study suggests the prognostic value of qEEG and explores the utility of combining neurophysiological monitoring with stereotactic aspiration surgery. The integration of these tools may assist in prognostic assessment for PBSH patients; however, validation in larger prospective studies is required before clinical adoption.

## Introduction

Primary brainstem hemorrhage (PBSH) is a severe neurological condition associated with a high all-cause mortality rate, accounting for 6% to 10% of all spontaneous intracerebral hemorrhages and representing one of the most devastating subtypes of hemorrhagic stroke [1,2]. Mortality rates for PBSH range from 47% to 80%, depending on the location and volume of bleeding [3]. Patients with small hematomas may remain conscious, presenting with symptoms such as dizziness, cranial nerve dysfunction, or limb weakness. In contrast, larger hematomas often result in severe manifestations, including coma, pupillary abnormalities, respiratory compromise, and paralysis [4,5]. These outcomes are largely influenced by the extent of hemorrhage, its anatomical location, and individual patient factors.

Despite its high mortality and morbidity, advancements in medical imaging, monitoring technologies, and minimally invasive surgical techniques have significantly improved treatment options for PBSH [2,6,7]. Among these innovations, stereotactic aspiration surgery has emerged as a promising approach for hematoma evacuation. This technique utilizes precise stereotactic guidance to locate hematomas, followed by minimally invasive aspiration through a burr hole [8]. It is characterized by reduced trauma to surrounding brain tissue, shorter ICU stays, and lower healthcare costs. Research has demonstrated its ability to improve patient outcomes and reduce secondary injury, establishing its feasibility for broader clinical adoption [9].

In addition to surgical advancements, neurophysiological monitoring techniques such as quantitative electroencephalography (qEEG) and transcranial Doppler (TCD) have proven valuable in managing patients with intracerebral hemorrhage [10,11]. qEEG quantifies electrical brain activity through advanced mathematical analysis, offering real-time, non-invasive monitoring of neural excitatory and inhibitory states. Its sensitivity in detecting changes in fast and slow brain waves makes it a critical tool for assessing brain function and predicting prognosis in patients undergoing stereotactic hematoma aspiration surgery.

TCD, another bedside monitoring technique, measures cerebral blood flow velocity and provides insights into cerebral hemodynamics, vasospasm, and autoregulation

[12]. By analyzing parameters such as pulsatility index (PI) and mean flow velocity (VM), TCD aids in evaluating surgical efficacy, detecting complications, and assessing neurological recovery. Its ability to offer continuous, non-invasive monitoring makes it an essential complement to qEEG in the postoperative care of PBSH patients.

This study investigates the combined application of qEEG and TCD monitoring in patients undergoing stereotactic hematoma aspiration surgery for PBSH. By integrating these two modalities, we aim to comprehensively assess cerebral hemodynamics and electrocortical activity, providing clinicians with real-time, actionable data to guide personalized treatment strategies. The findings of this study may contribute to improving long-term outcomes and advancing predictive models for PBSH management.

## Materials and methods

### Ethical approval and participant consent

This study was approved by the Ethics Review Committee of the Quzhou Affiliated Hospital of Wenzhou Medical University (approval No. 2018−071). The study adhered to the Declaration of Helsinki and relevant national regulations. Informed consent was obtained from all participants or their legally authorized representatives prior to inclusion. The individual pictured has provided written informed consent to publish their image alongside the manuscript.

### Clinical data

This retrospective observational study aimed to assess changes in brain function among patients with PBSH treated with stereotactic hematoma aspiration surgery compared to non-operative management. To ensure sample homogeneity and data reliability, strict inclusion and exclusion criteria were applied. Eligible subjects were adults (≥18 years old) diagnosed with PBSH between December 2022 and October 2023 at the Neurosurgery Department of Quzhou Hospital, affiliated with Wenzhou Medical University. Written informed consent was obtained from all participants for study participation.

Patients in the stereotactic surgery group underwent hematoma aspiration within 24–48 hours of symptom onset and received continuous quantitative qEEG and TCD monitoring for at least 4 hours within 24 hours post-surgery. Patients in the non-surgical group underwent at least one long-term qEEG and TCD monitoring session within 24 hours of admission. Inclusion criteria required complete medical records and monitoring data. Exclusion criteria ruled out individuals with severe comorbid brain diseases, surgical complications, incomplete monitoring data, severe systemic conditions, pregnancy, significant mental illnesses, or a history of substance abuse.

A total of 34 cases were initially identified and screened using pre-established exclusion criteria. Specific exclusions included: 1 patient with epilepsy, which could influence baseline brain electrical activity; 2 patients with Alzheimer's disease, whose neurodegeneration might confound brain function monitoring; 3 patients with cerebellar hemorrhage, differing pathophysiologically from PBSH; 3 cases with hematoma volumes under 5 mL and Glasgow Coma Scale (GCS) scores of 8 or above; 1 patient with hydrocephalus, potentially altering intracranial pressure and hemodynamics; 1 patient with postoperative intracranial infection, which might have affected monitoring indicators unrelated to the hemorrhage; and 1 patient with incomplete qEEG and TCD data. After rigorous screening, 25 patients met the study criteria: 16 underwent stereotactic hematoma puncture and aspiration, while 9 received non-operative management (Fig 1).

### Treatment methods

In the non-surgical group, the early management strategy primarily focused on maintaining airway patency to ensure unobstructed respiratory function. For patients with impaired respiratory function, the medical team promptly performed endotracheal intubation, supplemented by ventilator support when necessary, and considered early tracheostomy to minimize respiratory complications. Concurrently, proactive measures were taken to treat pulmonary infections, control inflammation, and adjust treatment plans to stabilize blood pressure and reduce cardiac workload. Additionally, for central

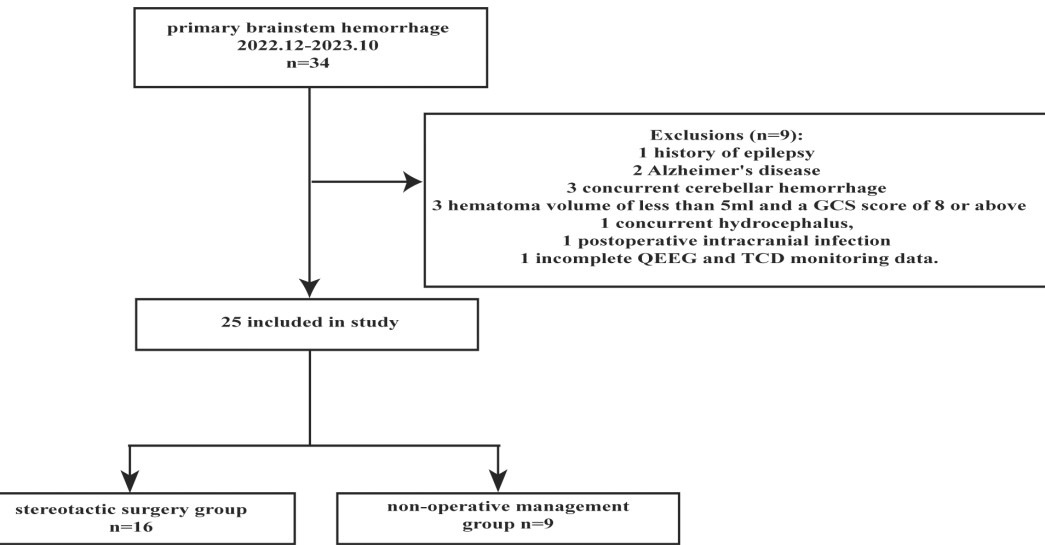

**Fig 1. Flowchart illustrating the patient inclusion process in the study.**

hyperthermia caused by brainstem hemorrhage, the team closely monitored body temperature and applied intermittent physical cooling as needed.

In the stereotactic aspiration and drainage surgery group, procedures were performed on patients with stable circulatory conditions, GCS scores below 8, hematoma volumes exceeding 3 mL, and with informed consent from the patients' families. During the surgery, the ASA-602S stereotactic head frame and ASA-620 surgical planning system (Anke, Shenzhen, China) were utilized to achieve precise puncture, aspiration, and drainage. Patients were intubated prior to surgery to prevent aspiration. The scalp was carefully prepared, and the base ring of the stereotactic head frame was installed with the patient in a prone position. Sedatives were administered during the procedure to alleviate discomfort and prevent severe coughing. The base ring was meticulously positioned so that its lower edge aligned with the angle of the mandible, ensuring full exposure of the posterior cranial fossa and sufficient space for surgical operations.

Subsequently, a CT scan was conducted to precisely locate the hematoma, and the collected data was imported into the surgical planning system. Based on the hematoma's specific morphology, its long axis was selected as the optimal puncture path. The X, Y, Z coordinates of the target, along with the ring and arc angles, were determined. Under general anesthesia, the stereotactic head frame was reinstalled, and the evacuation needle was slowly rotated and inserted along the pre-planned path until it reached the hematoma. Aspiration was performed using a 5 mL syringe, maintaining negative pressure below 2 mL to minimize the risk of brainstem injury while maximizing hematoma removal. To enhance hematoma clearance, repeated lavage of the cavity was conducted using normal saline. After completing the removal, a DE-106 drainage tube (SOPHYSA, France) was placed in the hematoma cavity.

Entry Zones for Brainstem Hemorrhage: Stereotactic routes for brainstem hematoma were stratified by anatomical level, including the midbrain, pons, and medulla, and by the relationship of the hematoma to the brainstem surface as dorsal, lateral, or ventral. The goal was to achieve the shortest feasible path while avoiding critical nuclei and long tracts. At the midbrain level, a lateral mesencephalic sulcus corridor was selected for lateral or dorsolateral midbrain hematomas, which avoids ventral passage through the densely packed corticospinal tract region and can be reached via a suboccipital posterolateral fossa route or by direct stereotactic access. A dorsal quadrigeminal plate route through the supracerebellar infratentorial corridor was reserved for shallow dorsal lesions, with strict depth control and avoidance of the midline. At the pontine level, a lateral pontine peritrigeminal entry zone around the trigeminal root entry zone was used for lateral or

lateroposterior hematomas, benefiting from distance to the facial colliculus and the medial longitudinal fasciculus. A middle cerebellar peduncle approach was employed for lateral pontine lesions or those extending into the peduncle, where the fiber architecture is relatively loose and conducive to establishing a working channel. For shallow dorsal lesions, a dorsal trans fourth ventricle route was used, with strict avoidance of the facial colliculus and entry confined to the suprafacial or infrafacial triangles. At the medullary level, a preolivary sulcus or lateral olivary corridor was considered for a small number of shallow ventrolateral lesions, with care to avoid the hypoglossal rootlets and the pyramids. For shallow dorsal lesions, provided a safe distance from the inferior medullary velum and the dorsal column nuclei could be maintained, a posterior median sulcus or a posterolateral sulcus entry was chosen (Table 1).

Postoperatively, a follow-up CT scan was performed to assess the clearance rate and overall surgical outcomes. If residual hematoma volume exceeded 3 mL, urokinase was administered to aid further evacuation. The use of urokinase for intracavitary clot lysis is off-label, and prior clinical studies have supported its reasonableness: it accelerates the resolution of intracranial hematomas and, in selected cohorts, when standardized protocols are followed, improves clinical outcomes without excess symptomatic hemorrhage. Drawing on representative trials and systematic reviews in spontaneous intracerebral hemorrhage and intraventricular hemorrhage that used similar dosing ranges [13,14], we instilled 20,000 IU diluted in 2 mL of preservative-free normal saline into the hematoma cavity, followed by a 1–2 mL saline flush; the catheter was clamped for 30–60 minutes and then reopened for drainage. Dosing was repeated every 12 hours for up to 3–5 days, with a maximum cumulative dose of 120,000–200,000 IU, and was stopped earlier if CT showed adequate evacuation or if rebleeding was suspected. With extensive experience and advanced surgical techniques, a high initial hematoma clearance rate was achieved, with only one patient requiring urokinase for additional treatment. Typically, the drainage tube was safely removed within 1–3 days after surgery.

## Monitoring

All patients underwent at least 4 hours of (Solar, Beijing, China) and TCD (VIASONIX, USA) monitoring within 24 hours of admission. For surgical patients, additional monitoring was performed within 24 hours post-operation. During the monitoring process, patients were positioned supine with the head of the bed elevated by 15–30 degrees to optimize conditions. For TCD monitoring, a 2-MHz pulsed-wave Doppler probe was applied. This study focused on monitoring the basilar artery, maintaining a depth of 80–100 millimeters (Fig 2).

qEEG monitoring followed the international 10–20 system, employing eight channels (Fp1, Fp2, C3, C4, T3, T4, O1, O2), with Fz as the ground electrode and Cz, A1, and A2 as reference electrodes. The EEG signal acquisition settings

**Table 1. Candidate entry zones for brainstem hemorrhage.**

| | Typical indication | Key landmarks used | Must-avoid structures | Default priority |
|---|---|---|---|---|
| **Midbrain** | Lateral/dorsolateral midbrain hematoma | LMS line vs peduncular contour | CST ventrally, CN III fascicles | High (if lateral) |
| **Peritrigeminal (pons)** | Lateral/lateroposterior pontine hematoma | CN V REZ vicinity, petrous landmarks | Facial colliculus/MLF medially | High |
| **Middle Cerebellar Peduncle(pons)** | Lateral pontine extending to MCP | MCP external contour on CT | AICA loops, deep tracts | High |
| **Trans–fourth ventricle (pons, dorsal)** | Shallow dorsal pontine | 4th ventricle floor curvature | Facial colliculus | Conditional (shallow only) |
| **Preolivary sulcus (medulla)** | Shallow ventrolateral medulla | Pyramid–olive groove | CN XII rootlets, pyramids | Low (selected cases) |

LMS, lateral mesencephalic sulcus; MCP, middle cerebellar peduncle; CST, corticospinal tract; CN, cranial nerve; REZ, root entry zone; MLF, medial longitudinal fasciculus; AICA, anterior inferior cerebellar artery; CT, computed tomography.

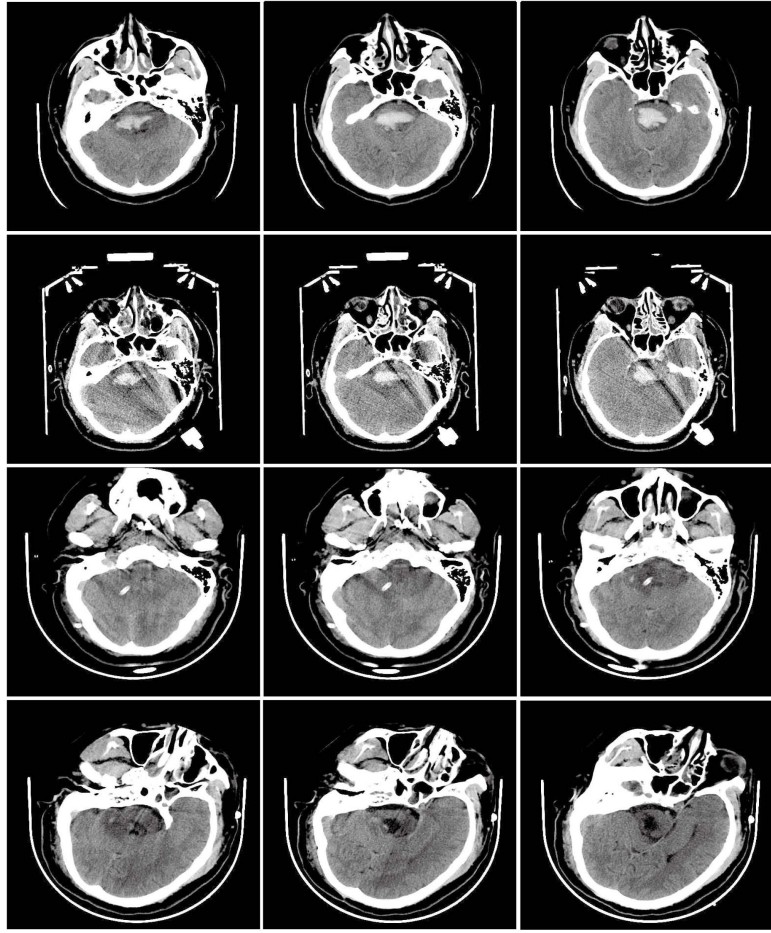

**Fig 2. Preoperative and postoperative assessments in patients with brainstem hemorrhage.** (A) Preoperative head CT scan showing the hematoma location. (B) Preoperative TCD monitoring of the basilar artery. (C) Preoperative quantitative electroencephalography (qEEG) monitoring.(D) Postoperative head CT scan showing hematoma evacuation. (E) Postoperative transcranial Doppler (TCD) monitoring of the basilar artery. (F) Postoperative quantitative electroencephalography (qEEG) monitoring.

aincluded a time base of 30 mm/s, sensitivity of 10 μV/mm, impedance maintained below 10 kΩ, a sampling frequency of 250 Hz, and bandpass filtering between 0.3 Hz and 30 Hz. Monitoring sessions lasted for a minimum of 4 hours, ensuring the acquisition of clear and stable signals for subsequent detailed analysis (Fig 3).

## Clinical data and data analysis

All patients received routine vital sign monitoring and intensive care in the neurocritical care unit and were observed for at least 90 days following brainstem hemorrhage. Clinical outcomes were assessed through hospital records, outpatient follow-ups, and telephone interviews. The survival status at 30 days (mortality) and neurological recovery at 90 days post-surgery were analyzed. Neurological function was evaluated using the modified Rankin Scale (mRS), which defines the following scores: asymptomatic (mRS 0), symptomatic without disability (mRS 1), mild disability (mRS 2), moderate disability but independently ambulatory (mRS 3), severe disability (mRS 4), bedridden (mRS 5), and death (mRS 6). For statistical analysis, patients were categorized into two groups: the "functional recovery" group (mRS 0–3) and the "non-functional recovery" group (mRS 4–6).

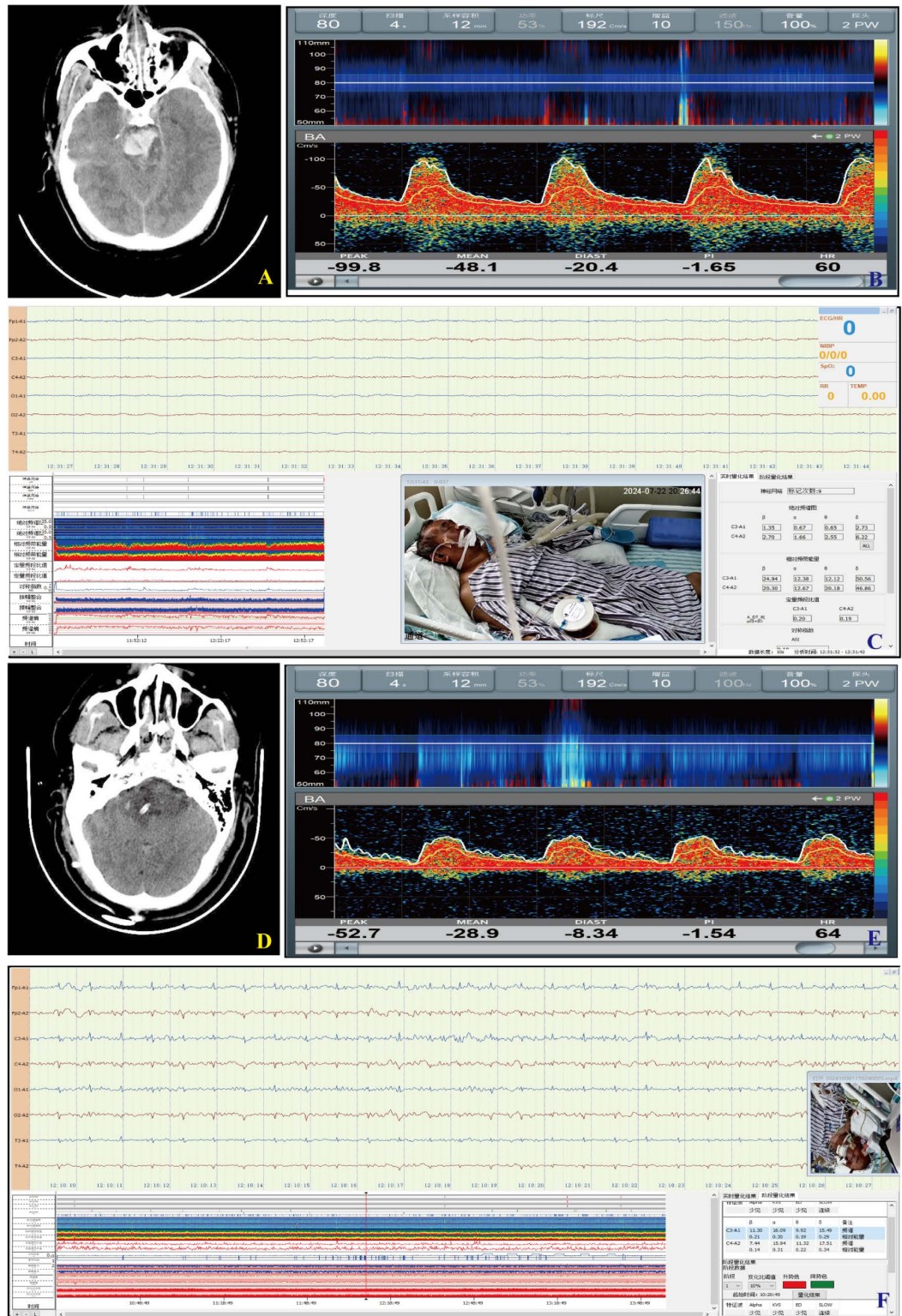

**Fig 3. Monitoring setup and data analysis of brain function in PBSH patients.** (A). Placement of EEG electrodes according to the international 10-20 system. (B). Raw electroencephalogram (EEG) recording showing brain activity. (C). Amplitude-integrated EEG (aEEG) pattern illustrating brain function post-surgery. (D). Relative band power (RBP) analysis showing energy distribution in different frequency bands.

## qEEG

Detailed offline quantitative analysis of EEG data was conducted using MATLAB tools (MathWorks, Natick, Massachusetts, USA). A 30-minute artifact-free segment of EEG data was selected for analysis. Spectral power for each electrode was calculated using the Fast Fourier Transform. To generate power spectral density maps, the Welch averaged periodogram method was applied, using a 2-second Hamming window with 50% overlap, ensuring a frequency resolution of 0.5 Hz. Absolute band power was calculated for four frequency bands: δ (1–3 Hz), θ (4–7 Hz), α (8–13 Hz), and β (14–30 Hz). Relative band power for each frequency band was determined by dividing the absolute power of the respective band by the total power within the 1–30 Hz range, yielding RBP δ%, RBP θ%, RBP α%, RBP β% and aEEG.

## TCD

TCD waveforms were analyzed to assess cerebral blood flow dynamics, including systolic velocity (Vs), representing the peak blood flow during heart contraction; diastolic velocity (Vd), indicating the lowest flow speed during diastole; and mean velocity (Vm), calculated as (Vs – Vd)/ 3 + Vd. The pulsatility index (PI), derived as (Vs – Vd)/ Vm, was used to evaluate vascular resistance and pulsatility. These parameters provided valuable insights into cerebral hemodynamics and vascular status.

## Statistical analysis

Continuous data were presented as mean ± standard deviation (SD) or median with interquartile range (IQR), depending on normality. The Shapiro-Wilk test was used to assess data distribution. Paired t-tests or Wilcoxon signed-rank tests were applied for comparing preoperative and postoperative data. Between-group comparisons were conducted using independent t-tests or Mann-Whitney U tests for continuous variables, and chi-square tests for categorical variables. Correlation analyses were performed using Spearman's rank correlation coefficient (ρ) to examine relationships between TCD/qEEG parameters and 90-day mRS scores.

Receiver Operating Characteristic (ROC) curve analysis was employed to evaluate the predictive performance of hematoma volume, qEEG-derived parameters (e.g., RBP δ%, RBP α%), and aEEG patterns. The area under the curve (AUC), sensitivity, specificity, and cutoff values were calculated for individual and combined predictive models. Statistical significance was set at a p-value < 0.05. All analyses were conducted using SPSS 21.0 software (IBM, Chicago, IL, USA).

## Result

### Baseline characteristics and clinical outcomes

As presented in Table 2, no statistically significant differences were observed in the baseline characteristics between the stereotactic surgery group (n = 16) and the non-surgical group (n = 9). Specifically, the mean age was 61.25 ± 12.60 years in the stereotactic surgery group and 49.22 ± 15.25 years in the non-surgical group (p = 0.06). Preoperative GCS scores were comparable between the two groups (4.38 ± 1.15 vs. 4.78 ± 1.20, p = 0.43), as were the mean hematoma volumes (10.10 ± 2.37 mL vs. 9.56 ± 1.90 mL, p = 0.54).

Further analysis of the treatment outcomes in the stereotactic aspiration surgery group revealed that hematomas were almost completely evacuated immediately after surgery in 11 patients, as confirmed by computed tomography (CT) scans (Fig 4). Only one patient in this group required additional urokinase treatment postoperatively to facilitate hematoma dissolution. We did not observe symptomatic rebleeding. However, given the small sample and nonrandomized design, no inferences about safety can be drawn.

There was no significant difference in the 30-day survival rate between the stereotactic surgery group and the non-surgical group (p = 0.36). However, a significant difference was observed in the modified Rankin Scale (mRS) scores at 90 days

**Table 2. Presents a comparative analysis of clinical characteristics, survival outcomes, neurological recovery, and prognostic indicators between the two patient cohorts.**

| Clinical data | Stereotactic surgery (n = 16) | Non-operative management (n = 9) | Test statistic | P |
|---|---|---|---|---|
| Sex[Case(%)] | | | 0[a] | 1.00 |
| Male | 14(87.50) | 8(88.89) | | |
| Female | 2(12.50) | 1(11.11) | | |
| Age[(mean±SD) years] | 61.25±12.60 | 49.22±15.25 | 2.01[b] | 0.06 |
| Preoperative GCS score | 4.38±1.15 | 4.78±1.20 | 0.82[b] | 0.43 |
| Heamatoma volume[(mean±SD) ml] | 10.10±2.37 | 9.56±1.90 | 0.63[b] | 0.54 |
| Survival status after 30d | | | 0.83[a] | 0.36 |
| Survival [Case(%)] | 13 (81.25) | 5 (55.56) | | |
| Death [Case (%)] | 3 (18.75) | 4 (44.44) | | |
| mRS score after 90 days | | | 5.53[a] | 0.01 |
| (mRS score 0–3) [Case (%)] | 11(68.75) | 1(11.11) | | |
| (mRS score 4–6) [Case (%)] | 5(31.25) | 8(88.89) | | |

Note: GCS, Glasgow Coma Scale; mRS, modified Rankin Scale, [a]$\chi^2$ value; [b]t value.

(p = 0.01). In the stereotactic surgery group, 68.75% of patients achieved favorable outcomes (mRS score 0–3), whereas only 11.11% of patients in the non-surgical group reached this outcome. Conversely, poor outcomes (mRS score 4–6) were more prevalent in the non-surgical group (88.89%) compared to the stereotactic surgery group (31.25%).

## TCD and qEEG parameters

**TCD parameters.** As presented in Table 3, significant changes in TCD parameters were observed after surgery. VS decreased from 95 (86, 105) cm/s preoperatively to 85 (75.75, 91) cm/s postoperatively (p < 0.01), while VM also showed a reduction from 56.5 (50.25, 63.5) cm/s to 52 (47.75, 58) cm/s (p < 0.01). PI decreased significantly from 1.19 (1.03, 1.31) to 0.92 (0.85, 1.03) (p < 0.01), indicating improved vascular elasticity and reduced vascular resistance. In contrast, no significant changes were observed in VD (p = 0.93).

**qEEG parameters.** qEEG analysis revealed significant postoperative changes in RBP parameters. The RBP δ decreased significantly from 55.31±7.09% to 23.44±8.59% (p < 0.01), while the RBP α increased from 13.81±3.49% to 38.88±10.51% (p < 0.01). No significant changes were observed in RBP θ and RBP β proportions (p = 0.08 and p = 0.27, respectively).

## Amplitude integrated electroencephalogram (aEEG) classification

aEEG classifications showed a significant shift postoperatively (p = 0.003). The proportion of patients classified as "normal" increased from 12.5% preoperatively to 50% postoperatively, while those classified as having "severe injury" decreased from 62.5% to 18.75%. The proportion of patients with "mild injury" remained relatively stable (25.0% preoperatively vs. 31.25% postoperatively).

## Correlation analyses

The correlation analysis (Table 4) identified several significant associations between clinical variables, TCD/qEEG parameters, and 90-day mRS scores. Hematoma volume exhibited a significant positive correlation with the 90-day mRS score (ρ = 0.497, p = 0.011), indicating that larger hematoma volumes were associated with poorer outcomes. Conversely,

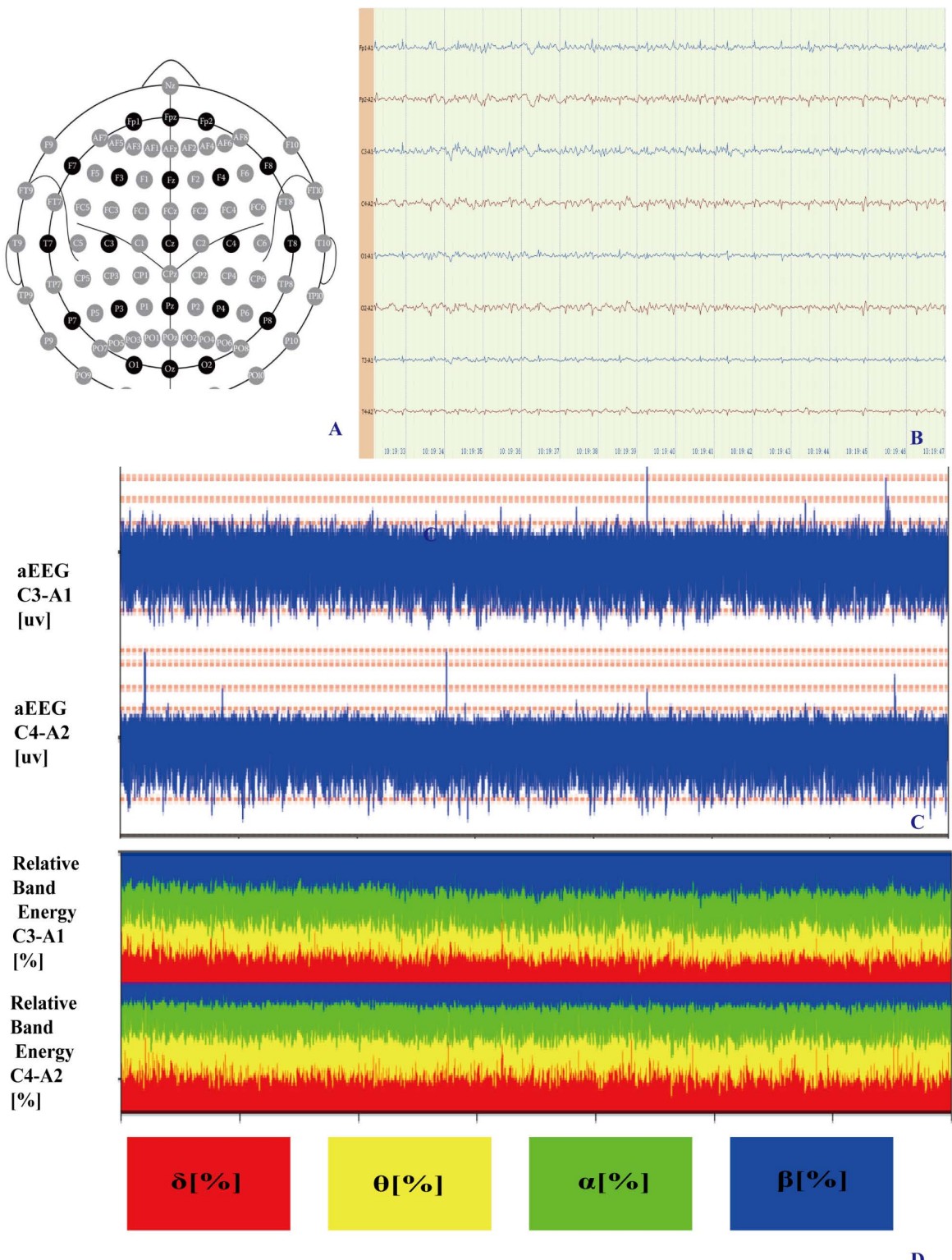

**Fig 4. CT scans of a patient with brainstem hemorrhage at different stages of treatment.** A-C: Preoperative head CT scans showing the initial hematoma. D-F: Post-implantation CT scans after the customized stereotactic frame placement. G-I: Immediate postoperative CT scans showing hematoma evacuation and brainstem status. J-L: Post-drainage tube removal CT scans, confirming hematoma clearance and brain recovery.

**Table 3. Transcranial Doppler and quantitative electroencephalography parameters.**

| | Preoperative | Postoperative | P value |
|---|---|---|---|
| **TCD Parameters** | | | |
| VS,cm/s | 95 (86, 105) | 85 (75.75, 91) | **<0.01** |
| VM,cm/s | 56.5 (50.25, 63.5) | 52 (47.75, 58) | **<0.01** |
| VD,cm/s | 35.5 (30.25, 42) | 37 (32.75, 42) | 0.93 |
| PI | 1.19 (1.03, 1.31) | 0.92 (0.85, 1.03) | **<0.01** |
| **qEEG Parameters** | | | |
| RBP δ% | 55.31±7.09 | 23.44±8.59 | **<0.01** |
| RBP θ% | 12.25±5.12 | 15.19±6.31 | 0.08 |
| RBP α% | 13.81±3.49 | 38.88±10.51 | **<0.01** |
| RBP β% | 18.62±5.69 | 20±10 | 0.27 |
| **aEEG** | | | |
| Normal(%) | 2(12.5%) | 8(50%) | **0.003** |
| Mild injury(%) | 4(25.0%) | 5(31.25%) | |
| Severe injury(%) | 10(62.5%) | 3(18.75%) | |

TCD: transcranial Doppler; qEEG: quantitative electroencephalography; VS: systolic flow velocity; VM: mean flow velocity; VD: diastolic flow velocity; PI: pulsatility index; RBP: Relative Band Power; aEEG: amplitude-integrated electroencephalography.

**Table 4. Clinical variables, TCD parameters, and qEEG parameters and their Spearman correlation with 90-day mRS.**

| Variable | Spearman Correlation | P value |
|---|---|---|
| **Clinical Variables** | | |
| Heamatoma volume | 0.497 | **0.011** |
| Surgery | −0.511 | **0.009** |
| Admission GCS | 0.078 | 0.712 |
| **TCD Parameters** | | |
| VS | 0.253 | 0.221 |
| VM | 0.180 | 0.390 |
| VD | −0.135 | 0.518 |
| PI | 0.282 | 0.171 |
| **QEEG Parameters** | | |
| RBPδ | 0.480 | **0.015** |
| RBPθ | 0.386 | 0.056 |
| RBPα | −0.456 | **0.022** |
| RBPβ | −0.234 | 0.261 |
| aEEG | 0.544 | **0.004** |

TCD, transcranial Doppler; qEEG, quantitative electroencephalography; GCS, Glasgow Coma Scale on admission; VS, systolic flow velocity; VM, mean flow velocity; VD, diastolic flow velocity; PI, pulsatility index; RBP, relative band power; aEEG, amplitude-integrated EEG.

surgical intervention showed a significant negative correlation with the 90-day mRS score ($\rho = -0.511$, $p = 0.009$), suggesting that surgery played a beneficial role in improving functional recovery.

None of the TCD parameters (VS, VM, VD, PI) demonstrated statistically significant correlations with the 90-day mRS score ($p > 0.05$). However, qEEG parameters revealed important insights: RBP δ power was significantly positively correlated

with the 90-day mRS score (ρ = 0.480, p = 0.015), indicating that elevated δ power was linked to worse outcomes, whereas RBP α power was significantly negatively correlated (ρ = −0.456, p = 0.022), suggesting that higher α power was associated with better neurological recovery. Furthermore, the aEEG pattern showed a strong positive correlation with the 90-day mRS score (ρ = 0.544, p = 0.004), underscoring its potential as a reliable predictive marker for long-term neurological outcomes.

### ROC curves

To evaluate the predictive performance of qEEG parameters and aEEG in patients with brainstem hemorrhage, several analyses were conducted. Maximum hematoma volume was recorded for all patients as a baseline clinical variable. For the non-surgical group, preoperative qEEG parameters, including RBP in the δ and α frequency bands, were collected. In the surgical group, postoperative qEEG parameters were analyzed to assess the impact of stereotactic aspiration surgery on neurophysiological activity. Both RBP δ and α power, as well as aEEG patterns, were evaluated for their ability to predict functional outcomes, which were categorized into two groups: favorable outcomes (mRS score 0–3) and poor outcomes (mRS score 4–6).

ROC curve analysis was performed to assess the predictive accuracy of these variables for the binary classification of 90-day mRS scores. The AUC values for individual parameters were as follows: hematoma volume (AUC = 0.718, 95% CI: 0.508–0.928), RBP δ% (AUC = 0.737, 95% CI: 0.533–0.941), RBP α% (AUC = 0.769, 95% CI: 0.571–0.967), and aEEG (AUC = 0.795, 95% CI: 0.627–0.962). Notably, the combined model integrating hematoma volume, RBP α%, and aEEG achieved the highest predictive performance with an AUC of 0.865 (95% CI: 0.723–1.000), indicating superior accuracy compared to individual parameters (Fig 5).

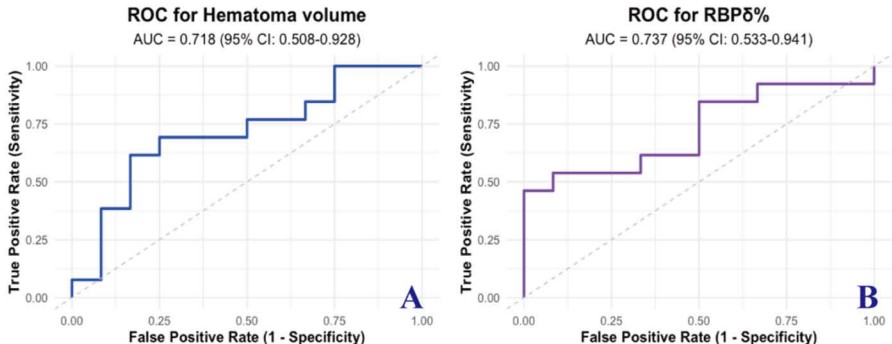

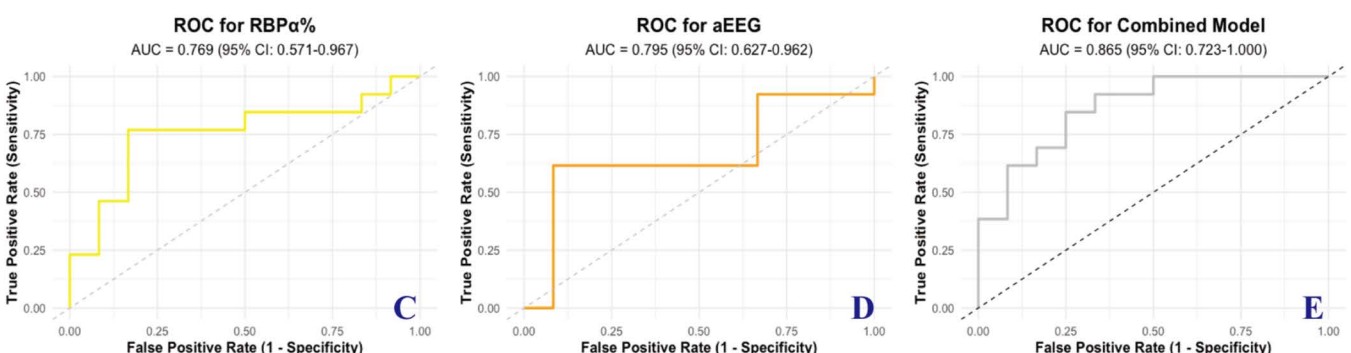

**Fig 5. ROC curves for predicting the mRS score at 90 days using various models.** (A) Hematoma volume (AUC = 0.718, 95% CI: 0.508–0.928); (B) RBP δ% (AUC = 0.737, 95% CI: 0.533–0.941); (C) RBP α% (AUC = 0.769, 95% CI: 0.571–0.967); (D) aEEG (AUC = 0.795, 95% CI: 0.627–0.962); (E) Combined model of Hematoma volume, RBP α%, and aEEG (AUC = 0.865, 95% CI: 0.723–1.000). AUC: Area Under the Curve; RBP: Relative Band Power; aEEG: Amplitude-Integrated Electroencephalography.

## Discussion

Our retrospective findings indicate a possible association between stereotactic aspiration surgery and improved functional outcomes relative to non-operative management, but confirmation in larger randomized or prospective studies is needed. Our findings are descriptive and hypothesis-generating. While some outcomes appeared numerically favorable in the stereotactic group, the study was not designed or powered to establish superiority over non-operative management. This is evidenced by the higher rates of favorable mRS scores observed at 90 days post-treatment. Additionally, this study highlights the utility of qEEG parameters and aEEG as predictive tools for functional outcomes in this patient population. Key predictors, including hematoma volume, qEEG-derived RBP δ and α power, and aEEG classifications, were significantly associated with 90-day mRS scores. Notably, the combined model integrating hematoma volume, RBPα%, and aEEG demonstrated superior predictive accuracy, achieving an AUC of 0.865, outperforming individual parameters. These findings underscore the importance of a multi-modal approach in prognostic assessments, offering a more comprehensive and accurate evaluation of patient outcomes.

Given the high level of trauma and technical complexity associated with craniotomy, stereotactic aspiration has has been proposed as a minimally invasive technique for intracerebral hemorrhage [3]. In the management of primary brainstem hemorrhage, this technique leverages the natural pathways formed by hematoma expansion and employs surgical trajectories carefully designed based on hematoma morphology to avoid critical neural tracts and densely populated nuclei regions. Comprehensive preoperative planning was conducted, and a 4-mm hematoma evacuation needle was utilized to precisely extract blood from the hematoma. This method ensures fixed-point targeting, accurate localization, and precise quantification of hematoma removal [15].

Stereotactic aspiration is performed without traction, electrocautery, or tissue separation, which has been proposed to reduce iatrogenic injury [16]. By promptly removing brainstem hematomas within 6–24 hours of onset, this technique effectively minimizes compression on intact brainstem structures and mitigates secondary damage to surrounding tissue. Early intervention also reduces the risk of postoperative rebleeding and further brainstem injury [8]. Among the 16 patients treated with this approach, no cases of postoperative rebleeding were observed, further supporting the safety and efficacy of this minimally invasive strategy.

The positive correlation between hematoma volume and mRS scores is consistent with prior reports of an association with poorer neurological outcomes [17]. Larger hematoma volumes, which can exert greater pressure on vital brainstem structures, were associated with worse functional recovery. Given the moderate predictive performance of hematoma volume alone (AUC = 0.718), adding neurophysiological markers may have value for prognostic assessment.

The significant positive correlation between RBP δ power and poor outcomes (ρ = 0.480) underscores the detrimental impact of δ wave dominance, which is often associated with deep cortical dysfunction and pathological brain states [18,19]. Conversely, the negative correlation between RBP α power and mRS scores (ρ = −0.456) reflects its association with cortical activation and functional recovery. These findings emphasize the value of QEEG in offering deeper insights into brain activity, complementing traditional clinical and imaging parameters in outcome prediction. The strong correlation between aEEG patterns and mRS scores (ρ = 0.544) further demonstrates its utility as a non-invasive tool for assessing neurological status. The postoperative shift in aEEG categories from "severe injury" to "normal" or "mild injury" highlights its sensitivity to treatment effects and recovery dynamics. Additionally, the higher AUC value for aEEG (0.795) compared to individual QEEG parameters suggests its potential as a standalone prognostic marker in this patient population. Notably, the combined model integrating hematoma volume, RBP α%, and aEEG demonstrated the highest predictive accuracy (AUC = 0.865), outperforming individual parameters. This finding underscores the importance of a multi-modal approach, as no single metric can fully capture the complexity of recovery processes in brainstem hemorrhage. By leveraging the complementary strengths of clinical, neurophysiological, and functional markers, the combined model establishes a more robust framework for personalized prognosis.

TCD parameters exhibited significant postoperative changes, such as reductions in VS and PI, no significant correlations were observed between these parameters and the 90-day modified Rankin Scale (mRS) scores. This lack of correlation highlights a key limitation of TCD: while it effectively measures hemodynamic changes in major cerebral arteries, it may not fully capture the microcirculatory and metabolic dynamics that are more closely associated with functional neurological recovery [20]. The primary role of TCD is to assess blood flow velocity and vascular resistance in large intracranial vessels, providing valuable insights into global cerebral perfusion and hemodynamic stability [21]. However, functional recovery in brainstem hemorrhage is likely influenced by more localized factors, such as microvascular perfusion, tissue oxygenation, and metabolic activity, which TCD cannot directly measure [22]. For instance, the brainstem contains densely packed neural tracts and nuclei that are highly sensitive to ischemia and metabolic disturbances [23]. These localized changes may play a more critical role in determining long-term outcomes than large-vessel hemodynamics alone [24]. Another consideration is the temporal nature of TCD measurements. While postoperative improvements in TCD parameters, such as reduced PI, may indicate enhanced vascular compliance and reduced resistance, they reflect an acute hemodynamic state rather than long-term recovery trajectories [25]. Functional outcomes like mRS scores are influenced by a combination of factors, including secondary injury processes, neuroplasticity, and the resolution of edema, which may not be directly mirrored in TCD findings. In conclusion, while TCD remains a valuable tool for monitoring large-vessel hemodynamics, its limitations in predicting functional outcomes in brainstem hemorrhage underscore the need for a multimodal approach. By integrating TCD with techniques that assess microvascular and metabolic factors, future studies could enhance its prognostic accuracy and provide deeper insights into the mechanisms underlying recovery.

The strengths of this study include the integration of clinical, neurophysiological, and functional assessments, and the use of ROC analysis to quantify predictive performance. However, several limitations should be acknowledged: The relatively small sample size may limit the generalizability of findings and reduce statistical power for subgroup analyses. The use of postoperative qEEG in the surgical group versus preoperative qEEG in the non-surgical group introduces potential variability. Single-center design may not account for institutional differences in treatment protocols. In conclusion, despite its limitations, this study provides a valuable framework for integrating clinical, neurophysiological, and functional assessments in the prognostic evaluation of brainstem hemorrhage. Future research should aim to address these limitations through larger, multicenter studies and more standardized data collection protocols, while exploring novel approaches to integrate dynamic and mechanistic insights into outcome prediction models.

Future research should focus on validating these findings in larger, multicenter cohorts to enhance generalizability and ensure robustness across diverse clinical settings. Additionally, studies should explore the dynamic evolution of qEEG and aEEG parameters over time to better capture recovery trajectories and identify critical time windows for intervention. Investigating the integration of advanced imaging techniques and biochemical markers with qEEG and aEEG could further develop a more comprehensive and accurate prognostic framework. Finally, evaluating the real-world applicability of the combined model in guiding clinical decision-making and optimizing individualized treatment planning will be essential to translate these findings into practice.

## Conclusion

This study highlights the prognostic value of qEEG and aEEG parameters in patients with brainstem hemorrhage and suggests that a multi-modal predictive approach may offer enhanced prognostic utility compared with single-parameter assessments. Additionally, our data indicate that stereotactic aspiration surgery is associated with reduced hematoma volume, alleviation of brainstem compression, and mitigation of secondary injury mechanisms. The integration of neurophysiological monitoring with surgical intervention shows promise for optimizing patient management strategies. While these observations require validation in prospective controlled studies, they provide a foundation for further exploration of personalized treatment protocols for brainstem hemorrhage.

## Acknowledgments

Thanks for the Kimi.ai for the English editing and academic writing.

## Author contributions

**Data curation:** Jin Wang.

**Funding acquisition:** Bingjie Jiang.

**Supervision:** Huayong Wu.

**Writing – original draft:** Bingjie Jiang.

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
