## [Decision Letter · Decision Letter 0]

20 Aug 2025

Dear Dr. Jiang,

Thank you for submitting your manuscript to PLOS ONE. After careful consideration, we feel that it has merit but does not fully meet PLOS ONE’s publication criteria as it currently stands. Therefore, we invite you to submit a revised version of the manuscript that addresses the points raised during the review process.

**ACADEMIC EDITOR REQUESTS:**

This is an interesting approach to a complex disease. Nonetheless, the manuscript will need some optimization prior to publication.

Major corrections:

Please describe the approaches taken to the brainstem and how often each was utilized. That is, which entry zone was used? If surgery was done under CT, how were you able to define the anatomy to specify the entry zone? Also, please give a better description of both the conventional and stereotactic approaches: a reader of your manuscript should be able to recreate what you've done after reading a scientific manuscript.

Please provide dosing for all drugs mentioned (e.g. urokinase). Additionally, since Urokinase is being used in an off-label fashion, this needs to be specified in the manuscript and justified using prior publications describing that use.

Statements such as " Importantly, no patients experienced complications such as rebleeding, further demonstrating the safety and efficacy of this surgical approach" or "The findings of this study demonstrate that stereotactic aspiration surgery provides significant advantages over conventional therapy in improving functional outcomes for patients with brainstem hemorrhage" are perhaps not justified points to make, given the data provided. The "safety" or "superiority" of this surgery should not commented on for several reasons: 1) the n is too small to make such claims, 2) not enough data/description is given to validate whether the treatment groups were properly balanced to allow for comparison: was only one surgeon involved in stereotactic approach vs one surgeon for conventional approach? Were multiple surgeons involved? What was the decision process to use one technique vs the other? Were patients randomized to treatment? If not, what decision making went into choosing one option vs another? Not enough detail is given. The authors are encouraged to describe the techniques and the outcomes without making values-based statements regarding safety or efficacy.

Minor corrections:

Line 464 "analyses. he use of postoperative qEEG in the" The authors likely mean "The use of" Please rigorously check the manuscript for typos and grammar errors before considering resubmitting.

We look forward to receiving your revised manuscript.

Kind regards,

Eric Anthony Sribnick, MD, PhD, FAANS

Academic Editor

PLOS ONE

Journal Requirements:

“This work was supported by the Key Scientific and Technological Project of Zhejiang Province, including the Quzhou City Guided Project (Grant No. 2021022 and Grant No. 2018086), as well as the Zhejiang Medical Association Clinical Research Fund Project (Grant No. 2022ZTC-A111).”

“This work was supported by the Key Scientific and Technological Project of Zhejiang Province, including the Quzhou City Guided Project (Grant No. 2021022 and Grant No. 2018086), as well as the Zhejiang Medical Association Clinical Research Fund Project (Grant No. 2022ZTC-A111).”

5. Please include your tables as part of your main manuscript and remove the individual files. Please note that supplementary tables (should remain/ be uploaded) as separate "supporting information" files".

6. We note that Figure 2 includes an image of a patient in the study.

Additional Editor Comments:

From the Academic Editor:

This is an interesting approach to a complex disease. Nonetheless, the manuscript will need some optimization prior to publication.

Major corrections:

Please describe the approaches taken to the brainstem and how often each was utilized. That is, which entry zone was used? If surgery was done under CT, how were you able to define the anatomy to specify the entry zone? Also, please give a better description of both the conventional and stereotactic approaches: a reader of your manuscript should be able to recreate what you've done after reading a scientific manuscript.

Please provide dosing for all drugs mentioned (e.g. urokinase). Additionally, since Urokinase is being used in an off-label fashion, this needs to be specified in the manuscript and justified using prior publications describing that use.

Statements such as " Importantly, no patients experienced complications such as rebleeding, further demonstrating the safety and efficacy of this surgical approach" or "The findings of this study demonstrate that stereotactic aspiration surgery provides significant advantages over conventional therapy in improving functional outcomes for patients with brainstem hemorrhage" are perhaps not justified points to make, given the data provided. The "safety" or "superiority" of this surgery should not commented on for several reasons: 1) the n is too small to make such claims, 2) not enough data/description is given to validate whether the treatment groups were properly balanced to allow for comparison: was only one surgeon involved in stereotactic approach vs one surgeon for conventional approach? Were multiple surgeons involved? What was the decision process to use one technique vs the other? Were patients randomized to treatment? If not, what decision making went into choosing one option vs another? Not enough detail is given. The authors are encouraged to describe the techniques and the outcomes without making values-based statements regarding safety or efficacy.

Minor corrections:

Line 464 "analyses. he use of postoperative qEEG in the" The authors likely mean "The use of" Please rigorously check the manuscript for typos and grammar errors before considering resubmitting.

Reviewers' comments:

Reviewer's Responses to Questions

**Comments to the Author**

1. Is the manuscript technically sound, and do the data support the conclusions?

Reviewer #1: No

Reviewer #2: Yes

2. Has the statistical analysis been performed appropriately and rigorously?

Reviewer #1: No

Reviewer #2: I Don't Know

3. Have the authors made all data underlying the findings in their manuscript fully available?

Reviewer #1: Yes

Reviewer #2: Yes

4. Is the manuscript presented in an intelligible fashion and written in standard English?

Reviewer #1: Yes

Reviewer #2: Yes

Reviewer #1: The sample size is small (n=25); a power analysis or justification is missing.

TCD parameters improved post-surgery but showed no correlation with mRS; explanation in discussion is limited.

Lack of external validation or multicenter data limits generalizability.

The timeframe between monitoring and outcome assessment (90 days) is appropriate but not clearly discussed in terms of neurological recovery stages.

Reviewer #2: nicely written article. It highlight an area where not much progress has been made yet. the results indicates more work necessary to improve existing treatment paradigms for primary brainstem hemorrhage. thanks

**Do you want your identity to be public for this peer review?** For information about this choice, including consent withdrawal, please see our Privacy Policy

Reviewer #1: No

Reviewer #2: No

---

## [Author Response · Author response to Decision Letter 1]

31 Aug 2025

“PONE-D-25-17756 Minimally Invasive Surgery and Neurophysiological Monitoring for Brainstem Hemorrhage: Advancing Predictive Models with qEEG and TCD

ACADEMIC EDITOR REQUESTS:

Major corrections:

1. Please describe the approaches taken to the brainstem and how often each was utilized. That is, which entry zone was used? If surgery was done under CT, how were you able to define the anatomy to specify the entry zone? Also, please give a better description of both the conventional and stereotactic approaches: a reader of your manuscript should be able to recreate what you've done after reading a scientific manuscript.

Response: We have added a new subsection “Entry Zones for Brainstem Hemorrhage” and” Table 1” in Methods that details the candidate entry zones and our selection algorithm.

2. Please provide dosing for all drugs mentioned (e.g. urokinase). Additionally, since Urokinase is being used in an off-label fashion, this needs to be specified in the manuscript and justified using prior publications describing that use.

Respouse: We have added dosing information for urokinase. “The use of urokinase for intracavitary clot lysis is off-label, and prior clinical studies have supported its reasonableness: it accelerates the resolution of intracranial hematomas and, in selected cohorts, when standardized protocols are followed, improves clinical outcomes without excess symptomatic hemorrhage. Drawing on representative trials and systematic reviews in spontaneous intracerebral hemorrhage and intraventricular hemorrhage that used similar dosing ranges, we instilled 20,000 IU diluted in 2 mL of preservative-free normal saline into the hematoma cavity, followed by a 1–2 mL saline flush; the catheter was clamped for 30–60 minutes and then reopened for drainage. Dosing was repeated every 12 hours for up to 3–5 days, with a maximum cumulative dose of 120,000–200,000 IU, and was stopped earlier if CT showed adequate evacuation or if rebleeding was suspected.”

3. Statements such as " Importantly, no patients experienced complications such as rebleeding, further demonstrating the safety and efficacy of this surgical approach" or "The findings of this study demonstrate that stereotactic aspiration surgery provides significant advantages over conventional therapy in improving functional outcomes for patients with brainstem hemorrhage" are perhaps not justified points to make, given the data provided. The "safety" or "superiority" of this surgery should not commented on for several reasons: 1) the n is too small to make such claims, 2) not enough data/description is given to validate whether the treatment groups were properly balanced to allow for comparison: was only one surgeon involved in stereotactic approach vs one surgeon for conventional approach? Were multiple surgeons involved? What was the decision process to use one technique vs the other? Were patients randomized to treatment? If not, what decision making went into choosing one option vs another? Not enough detail is given. The authors are encouraged to describe the techniques and the outcomes without making values-based statements regarding safety or efficacy.

Response We thank the reviewer for this important point.

• Original:“Importantly, no patients experienced complications such as rebleeding, further demonstrating the safety and efficacy of this surgical approach.”Revised to:“We did not observe symptomatic rebleeding. However, given the small sample and nonrandomized design, no inferences about safety can be drawn.”

• Original:“The findings of this study demonstrate that stereotactic aspiration surgery provides significant advantages over conventional therapy in improving functional outcomes for patients with brainstem hemorrhage.”Revised to:“Our findings are descriptive and hypothesis generating. While some outcomes appeared numerically favorable in the stereotactic group, the study was not designed or powered to establish superiority over conventional therapy.”

4. Minor corrections: Line 464 "analyses. he use of postoperative qEEG in the" The authors likely mean "The use of" Please rigorously check the manuscript for typos and grammar errors before considering resubmitting.

Response Thank you for the reminder; we have made the corresponding changes.

5. If applicable, we recommend that you deposit your laboratory protocols in protocols.io to enhance the reproducibility of your results. Protocols.io assigns your protocol its own identifier (DOI) so that it can be cited independently in the future.

Response Thank you for the suggestion. As this study depositing on protocols.io is not applicable. We have provided methodological details and parameters in the Methods.

Journal Requirements:

Response: The corresponding changes have been made.

2�Thank you for stating in your Funding Statement: “This work was supported by the Key Scientific and Technological Project of Zhejiang Province, including the Quzhou City Guided Project (Grant No. 2021022 and Grant No. 2018086), as well as the Zhejiang Medical Association Clinical Research Fund Project (Grant No. 2022ZTC-A111).” Please provide an amended statement that declares all the funding or sources of support (whether external or internal to your organization) received during this study, as detailed online in our guide for authors at http://journals.plos.org/plosone/s/submit-now. Please also include the statement “There was no additional external funding received for this study.” in your updated Funding Statement. Please include your amended Funding Statement within your cover letter. We will change the online submission form on your behalf.

Response: The corresponding changes have been made.

3. Thank you for stating the following financial disclosure:Please state what role the funders took in the study.

Response: The funders had the following involvement with the study: study design, decision to publish, and preparation of the manuscript.

Respouse: All relevant data are available from Figshare at https://doi.org/10.6084/m9.figshare.30018478.

5. Please include your tables as part of your main manuscript and remove the individual files. Please note that supplementary tables (should remain/ be uploaded) as separate "supporting information" files".

Response: The corresponding changes have been made.

6. We note that Figure 2 includes an image of a patient in the study.

Respouse: We have obtained written informed consent for publication using the PLOS consent form. We have revised the Methods and Ethics statement to explicitly state this and added a note to the Figure 2 legend. The signed consent form has been securely filed in the patient’s case notes and will not be submitted.

Review Comments

Reviewer #1: The sample size is small (n=25); a power analysis or justification is missing.TCD parameters improved post-surgery but showed no correlation with mRS; explanation in discussion is limited.Lack of external validation or multicenter data limits generalizability.The timeframe between monitoring and outcome assessment (90 days) is appropriate but not clearly discussed in terms of neurological recovery stages.

Response: Given the rarity of primary brainstem hemorrhage, our sample consists of consecutive cases accrued within a predefined period (n=25). TCD indices capture acute cerebrovascular hemodynamics, whereas mRS reflects global disability influenced by multiple factors such as baseline hematoma volume and location, initial severity, secondary complications, and rehabilitation. Additionally, mRS may exhibit floor/ceiling effects in severe brainstem injury, and any relationship may be non-linear or time-dependent. We agree. The single-center design limits generalizability. In our study, 90 day mRS was pre specified as the primary outcome to align with common practice in stroke research and to facilitate comparability across studies. A detailed staging framework of neurological recovery would require longitudinal clinical scales and serial assessments that were beyond the scope and data structure of the present work. To avoid speculation, we kept the discussion concise. No changes were made in response to this comment, but we would be happy to elaborate if the Editor deems it necessary.

Reviewer #2: nicely written article. It highlight an area where not much progress has been made yet. the results indicates more work necessary to improve existing treatment paradigms for primary brainstem hemorrhage.

Response: We thank Reviewer #2 for the positive feedback and encouragement. We agree that this is an area with limited progress, and our findings underscore the need for continued efforts to refine treatment paradigms for primary brainstem hemorrhage.

---

## [Editor Report · Decision Letter 1]

11 Sep 2025

Dear Dr. Jiang,

Thank you for submitting your manuscript to PLOS ONE. After careful consideration, we feel that it has merit but does not fully meet PLOS ONE’s publication criteria as it currently stands. Therefore, we invite you to submit a revised version of the manuscript that addresses the points raised during the review process.

We look forward to receiving your revised manuscript.

Kind regards,

Eric Anthony Sribnick, MD, PhD, FAANS

Academic Editor

PLOS ONE

Journal Requirements:

Additional Editor Comments:

Academic Editor:

As the reviewers mentioned before, the manuscript is quite interesting and has many positive attributes. However, before the manuscript can be accepted, there are still further changes that should be made.

1) It appears that patients receiving "conventional therapy" received only supportive care and no surgery. If that is correct, please strongly consider referring to "conventional therapy" as "non-operative management" to avoid confusion. Also, in the discussion, the "conventional group" is called the "non-surgical group." For the sake of simplicity, please pick one name for each group and use it consistently throughout.

2) Lines 348-350. Please either remove this or change it to say something like "There was no significant difference in mortality between the treatment groups." The 30 day survival rate showed no significant difference, so you should not present this as a change. You have defined statistical significance as p<0.05, so any change that does not meet this definition is not a true change and should not be elaborated on as such. Any other nonsignificant differences should be treated similarly.

3) Because this is retrospective, because you did not randomize treatment, because your sample size is quite small, please do not write that one treatment is superior to another. Lines 439-441 still read that "stereotactic aspiration provides significant advantages over conventional therapy" and your study is not robust enough to make this claim.

4) Again in the conclusion, reference is made to "the superiority" of a multi-modal approach. The study is not designed to address superiority, so I would remove that or significantly change the language. The next line reads that there are "clinical benefits" to stereotactic aspiration. At best, because this is a non-randomized retrospective study, you could say that the surgical approach with aspiration correlates with a lower modified Rankin scale. The line after that reads "improves prognostic accuracy" and "enhances functional outcomes." Again, please remove this language or significantly change it. The next line reads that "the findings support the incorporation . . . better long-term recovery in patients with brainstem hemorrhage. Again, please remove this language or significantly change it.

5) In the manuscript, there are multiple occurrences of this type of language (claims not supported by the data presented). Please change these.

If these suggestions are incompatible with the type of message you are trying to convey by your manuscript, this is perhaps not the correct journal for the manuscript.

---

## [Author Response · Author response to Decision Letter 2]

16 Sep 2025

Academic Editor comments:

1. It appears that patients receiving "conventional therapy" received only supportive care and no surgery. If that is correct, please strongly consider referring to "conventional therapy" as "non-operative management" to avoid confusion. Also, in the discussion, the "conventional group" is called the "non-surgical group." For the sake of simplicity, please pick one name for each group and use it consistently throughout.

Respond Thank you very much for your detailed and important suggestions regarding the terminology. We have made the corresponding changes in the manuscript and highlighted them in yellow.

2. Lines 348-350. Please either remove this or change it to say something like "There was no significant difference in mortality between the treatment groups." The 30 day survival rate showed no significant difference, so you should not present this as a change. You have defined statistical significance as p<0.05, so any change that does not meet this definition is not a true change and should not be elaborated on as such. Any other nonsignificant differences should be treated similarly.

Respond: Based on the reviewers' comments and standard practices for reporting statistical significance, we have made the following changes to this paragraph: "There was no significant difference in the 30-day survival rate between the stereotactic surgery group and the non-surgical group (p = 0.36). However, a significant difference was observed in the modified Rankin Scale (mRS) scores at 90 days (p = 0.01). In the stereotactic surgery group, 68.75% of patients achieved favorable outcomes (mRS score 0–3), whereas only 11.11% of patients in the non-surgical group reached this outcome. Conversely, poor outcomes (mRS score 4–6) were more prevalent in the non-surgical group (88.89%) compared to the stereotactic surgery group (31.25%)." and highlighted the changes in yellow.

3. Because this is retrospective, because you did not randomize treatment, because your sample size is quite small, please do not write that one treatment is superior to another. Lines 439-441 still read that "stereotactic aspiration provides significant advantages over conventional therapy" and your study is not robust enough to make this claim.

Respond Based on the comment and the limitations of our study design, we have revised the original sentence—“The findings of this study demonstrate that stereotactic aspiration surgery provides significant advantages over non-operative management in improving functional outcomes for patients with brainstem hemorrhage.”—to: “Our retrospective findings indicate a possible association between stereotactic aspiration surgery and improved functional outcomes relative to non-operative management, but confirmation in larger randomized or prospective studies is needed.” and highlighted the changes in yellow.

4. Again in the conclusion, reference is made to "the superiority" of a multi-modal approach. The study is not designed to address superiority, so I would remove that or significantly change the language. The next line reads that there are "clinical benefits" to stereotactic aspiration. At best, because this is a non-randomized retrospective study, you could say that the surgical approach with aspiration correlates with a lower modified Rankin scale. The line after that reads "improves prognostic accuracy" and "enhances functional outcomes." Again, please remove this language or significantly change it. The next line reads that "the findings support the incorporation . . . better long-term recovery in patients with brainstem hemorrhage. Again, please remove this language or significantly change it. In the manuscript, there are multiple occurrences of this type of language (claims not supported by the data presented). Please change these.

Respond: We agree with the reviewers’ comments and have revised the deterministic and causal terms “superiority,” “clinical benefits,” “improves/enhances,” and “support the incorporation.” The conclusion has been updated to:“This study highlights the prognostic value of qEEG and aEEG parameters in patients with brainstem hemorrhage and suggests that a multi modal predictive approach may offer enhanced prognostic utility compared with single parameter assessments. Additionally, our data indicate that stereotactic aspiration surgery is associated with reduced hematoma volume, alleviation of brainstem compression, and mitigation of secondary injury mechanisms. The integration of neurophysiological monitoring with surgical intervention shows promise for optimizing patient management strategies. While these observations require validation in prospective controlled studies, they provide a foundation for further exploration of personalized treatment protocols for brainstem hemorrhage.” and highlighted the changes in yellow.

5. In the manuscript, there are multiple occurrences of this type of language (claims not supported by the data presented). Please change these.

Respond: We agree with the reviewer’s comments. In the Abstract, the Background section was revised from “Stereotactic hematoma aspiration surgery offers an effective treatment, and postoperative brainstem function monitoring is crucial for optimizing patient outcomes.” to “Stereotactic hematoma aspiration surgery has been explored as a treatment option, and postoperative brainstem function monitoring is considered important for patient management.”

In the Abstract’s Results section, “Compared to non-surgical group, stereotactic surgery showed no significant differences in baseline characteristics but resulted in higher favorable outcomes at 90 days” was revised to “Stereotactic surgery was associated with higher rates of favorable outcomes at 90 days compared with the non-surgical group.”

In the Abstract’s Conclusion section, “This study demonstrates the prognostic value of qEEG and highlights the benefits of combining neurophysiological monitoring with stereotactic aspiration surgery. The integration of these tools improves prognosis and functional outcomes, supporting their routine use in clinical practice.” was revised to “This study suggests the prognostic value of qEEG and explores the utility of combining neurophysiological monitoring with stereotactic aspiration surgery. The integration of these tools may assist in prognostic assessment for PBSH patients; however, validation in larger prospective studies is required before clinical adoption.”

In the Discussion section of the main text, “Given the high level of trauma and technical complexity associated with craniotomy, stereotactic aspiration has emerged as a promising minimally invasive alternative for the treatment of intracerebral hemorrhage” was revised to “Given the high level of trauma and technical complexity associated with craniotomy, stereotactic aspiration has been proposed as a minimally invasive technique for intracerebral hemorrhage.”

Additionally, “Unlike conventional surgical approaches, stereotactic aspiration avoids the use of traction, electrocautery, or tissue separation during the procedure, significantly reducing the risk of iatrogenic injury” was revised to “Stereotactic aspiration is performed without traction, electrocautery, or tissue separation, which has been proposed to reduce iatrogenic injury.”

“The positive correlation between hematoma volume and mRS scores aligns with previous findings, reaffirming its role as a critical determinant of poor neurological outcomes. Larger hematoma volumes impose greater pressure on vital brainstem structures, resulting in worse functional recovery. However, the moderate predictive performance of hematoma volume alone (AUC = 0.718) highlights its limitations in providing accurate prognostic information, necessitating the incorporation of additional neurophysiological markers.” was revised to “The positive correlation between hematoma volume and mRS scores is consistent with prior reports of an association with poorer neurological outcomes. Larger hematoma volumes, which can exert greater pressure on vital brainstem structures, were associated with worse functional recovery. Given the moderate predictive performance of hematoma volume alone (AUC = 0.718), adding neurophysiological markers may have value for prognostic assessment.” and highlighted the changes in yellow.

---

## [Editor Report · Decision Letter 2]

18 Sep 2025

Minimally Invasive Surgery and Neurophysiological Monitoring for Brainstem Hemorrhage: Advancing Predictive Models with qEEG and TCD

PONE-D-25-17756R2

Dear Dr. Jiang,

We’re pleased to inform you that your manuscript has been judged scientifically suitable for publication and will be formally accepted for publication once it meets all outstanding technical requirements.

Kind regards,

Eric Anthony Sribnick, MD, PhD, FAANS

Academic Editor

PLOS ONE

Additional Editor Comments (optional):

none additional

---

## [Editor Report · Acceptance letter]

PONE-D-25-17756R2

PLOS ONE

Dear Dr. Jiang,

I'm pleased to inform you that your manuscript has been deemed suitable for publication in PLOS ONE. Congratulations! Your manuscript is now being handed over to our production team.

Kind regards,

on behalf of

Dr. Eric Anthony Sribnick

Academic Editor

PLOS ONE